# Association between Weight Status and Mental Health among Korean Adolescents: A Nationwide Cross-Sectional Study

**DOI:** 10.3390/children10040620

**Published:** 2023-03-25

**Authors:** Youngha Choi, Jeana Hong

**Affiliations:** 1Department of Pediatrics, Kangwon National University Hospital, Chuncheon 24289, Republic of Korea; younghachoi0104@gmail.com; 2Department of Pediatrics, Kangwon National University School of Medicine, Chuncheon 24289, Republic of Korea

**Keywords:** adolescent, mental health, body mass index, weight perception, obesity

## Abstract

This study explored the relationship between adolescents’ weight status and mental health problems. It specifically investigated the weight perceptions of obese adolescents and the effect on their mental health. This cross-sectional study was based on the data of adolescents aged 12–18 years from the Korean National Health and Nutritional Examination Survey (2010–2019). Data regarding anthropometric measurements, health conditions, and socioeconomic status were extracted, and the associations between weight status (actual, perceived, or misperceived) and mental health conditions (depressed mood, perceived stress, and suicidal ideation) were analyzed using complex sample multiple logistic regression after adjusting for possible confounders. A total of 5683 adolescents (53.1% boys and 46.9% girls) were included in this study, with a mean age of 15.1 years. Among the participants, actual, perceived, and misperceived status of being overweight were observed in 20.8%, 32.7%, and 18.4%, respectively. Additionally, depressed mood, perceived stress, and suicidal ideation were observed in 9.1%, 25.7%, and 7.4% of Korean adolescents, respectively, with higher prevalences in girls for all three conditions. Actual weight status was not significantly associated with mental health conditions in either sex. Furthermore, girls who perceived themselves to be overweight, regardless of their actual body weight, or who had overestimated their actual weight were more likely to have experienced depressed mood and stress, while boys who perceived themselves to be underweight were more likely to have experienced suicidal ideation than participants with an average weight perception or an accurate recognition of their weight status. Conversely, in overweight/obese participants, perceived weight status was not associated with mental health conditions. In conclusion, perceived weight status and its discrepancy with actual body weight were more strongly associated with an increased risk of mental health problems than actual weight status itself among Korean adolescents. Therefore, adolescents’ perceptions of their body image and weight-related attitude should be assessed to promote their mental health.

## 1. Introduction

Adolescence is a unique and formative period of development, encompassing biological, psychological, and social changes as individuals transition from childhood to adulthood [1]. Along with experiencing a growth spurt and hormonal changes during this period, adolescents undergo significant emotional and social-environmental changes [1,2]. As adolescence is a time of identity formation and increased independence, adolescents may experience increased emotional intensity and mood swings, and they may engage in greater risk-taking behaviors [3]. Moreover, they may become more involved in peer relationships and face social pressure to conform to norms related to appearance or behavior [4]. These factors can make them vulnerable to various mental health issues, including risk-taking behaviors, such as depression, anxiety, behavioral disorders, eating disorders, suicide, or substance use disorders [5,6,7].

In addition, childhood obesity is another growing concern worldwide that has been linked to various physical and psychological health problems [8,9]. Obese children are more likely to experience obesity stigma, teasing, or bullying by their peers, which in turn adversely affects their physical and emotional health [10]. Compared with their average-weight peers, obese children have higher rates of psychological problems, such as low self-esteem, poor health-related quality of life, depression, anxiety, and behavioral disorders [8,9,10]. Therefore, considering the possible negative consequences, the monitoring and screening of obese children’s mental health is highly recommended during the treatment of childhood obesity for timely psychiatric intervention [11]. However, previous studies evaluating the relationship between childhood obesity and mental health have shown variable results regarding the strength of the association according to their definitions of mental health problems, sex, and ethnic differences, suggesting a need for further investigation [12,13,14].

Several adult studies have found that both being underweight and severely obese were associated with an increased risk of depression, while the association with being overweight was less clear [15]. In other adult studies, weight perception—that is, perceived body image—has been suggested to be a more significant factor for mental health than actual weight status [16,17]. With the widespread adoption of social networking sites, adolescents are increasingly exposed to unrealistically thin and idealized body shapes, leading to increased body dissatisfaction and the development of unrealistic and distorted body ideals [18]. Several adolescent studies have reported that body image is more strongly associated with mental health than their actual weight [19,20]. However, it is unclear whether the findings reflect cultural and ethnic differences or the characteristics of adolescence. In addition, along with associations with mental health, gender differences have been reported; girls show a preference for a leaner body shape, whereas boys show a preference for a muscular body shape [21,22]. These findings highlight the need for further investigation into the relationships between weight status, body image distortion, and mental health among adolescents, while considering sex differences.

Therefore, this study aimed to determine whether actual weight, perceived weight status, and body image distortion were associated with mental health among Korean adolescents. Additionally, the study performed a subgroup analysis among obese adolescents to evaluate weight perceptions and their effect on obese adolescents’ mental health.

## 2. Materials and Methods

### 2.1. Study Design and Participants

In this study, we analyzed data from the Korean National Health and Nutrition Examination Survey (KNHANES), a national surveillance program designed to assess the health and nutritional status of non-institutionalized civilians in Korea. This nationwide, cross-sectional survey has been conducted every year since 2007 by the Korea Centers for Disease Control and Prevention (KCDC). It comprises three components: health interviews, health examinations, and nutrition surveys. The representative households who participate in the KNHNES are selected using a stratified and multi-stage clustered probability sampling design, where individuals aged 1 year and over are chosen for the survey. The health interviews, health examinations, and nutrition surveys are performed by a trained survey team, including health professionals, support staff, and dieticians, using face-to-face interview methods. During the health interviews, detailed information regarding the participants’ socioeconomic status, health-related behaviors, medical conditions, and mental health status is collected. During the course of the nutrition survey, the total amount of daily nutrients and calorie intake is calculated from the food intake questionnaire, which is designed as an open-ended survey using the 24 h recall method. A detailed description of the KNHANES design and data profile has been published as a separate paper [23].

We obtained data from adolescents aged 12–18 years from the KNHANES conducted from 2010 to 2019. Detailed information on the participants’ socioeconomic status, anthropometric measures, health conditions, and total daily calorie intake was extracted from the survey. Participants who did not complete the mental health questions in the health interviews were excluded from the final analysis.

### 2.2. Measurements

#### 2.2.1. Body Weight Status

During the health examination, the participant’s weight and height are measured, and their body mass index (BMI) is calculated and presented. We categorized the study population into four groups based on their BMI relative to their sex and age, according to the 2017 Korean National Growth Chart, as follows: underweight (below the 5th percentile), normal (from the 5th to the 85th percentile), overweight (from the 85th to the 95th percentile), and obese (above the 95th percentile) [24]. We assumed the categorized BMI variable as an indicator of the participant’s actual weight status. We utilized the participants’ responses to the question, “How do you feel about your body?”, which was posed in the health interview to assess their self-perceived weight status. Their possible answers were one of the following five options: “very thin”, “thin”, “average”, “fat”, and “very fat”. For analysis purposes, the responses were abridged into three categories, and we created the following new variables: “thin” (including “very thin” and “thin”), “average”, and “fat” (including “fat” and “very fat”), indicating the participants’ weight perceptions of being underweight, average, or overweight.

In addition, we created a variable to assess the participants’ misperceptions of their body weight. When the actual weight status of the participants, represented as a BMI percentile categorized into one of four groups, was matched to the four perceived weight status groups of very thin, thin combined with average, fat, and very fat, the participants were classified into a concordance group. When there was a discrepancy between their actual weight status and their perceived weight status, the participant was categorized into a discordant group, in which individuals who perceived their weight as being lower or higher than their actual body weight were further classified into an underestimated or overestimated discordant subgroup.

In addition, the participants were asked whether they had attempted to control their weight as a means to alter their body image during the preceding year. Participants who responded that they had attempted to control their weight, either by reducing it or by gaining weight, were classified into a group that had the intention to control their weight.

#### 2.2.2. Health Condition

Self-reported questionnaires from the health interview survey were used to evaluate the participants’ health status. Perceived health status was assessed by the question, “What do you think of your health in general?”, where the possible answers to choose from were “very good”, “good”, “fair”, “bad”, and “very bad.” For analysis, we categorized the responses into three groups: “very good, good”, “fair”, and “bad, very bad”. 

The information regarding mental health was obtained from the health interviews. Participants who answered yes to the question, “During the past year, have you ever felt so sad or desperate that it interfered with your daily life for at least two consecutive weeks?”, were classified as having experienced a depressive mood. We used the participants’ responses to the question, “How much stress do you feel in your daily life?”, to classify participants who responded “very much” or “much” into the perceived stress group, in contrast with those participants who answered “mild” or “none.” Suicidal ideation was assessed with the question, “In the last year, did you think about committing suicide?” A “yes” or “no” response was also used to determine whether the participants had contemplated suicide during the preceding year.

#### 2.2.3. Socio-Demographic Variables

Data on socioeconomic characteristics, such as sex, household income, and residential area, that could be associated with the participants’ weight or mental health status were obtained. The household income of the study population, classified by quartiles, was divided into three groups of equalized household income: low (<25 percentile), middle (25–75 percentile), and high (≥75 percentile). Areas of residency were categorized into urban and rural areas. Participants’ total daily calorie intake was extracted from the nutrition survey as one of the adjusting variables for the analysis.

### 2.3. Statistical Analyses

To represent the Korean population, the sample weights were generated by considering complex survey design, non-response rate, and post-stratification, which applied to all analyses. The complex sample Rao-Scott χ^2^ test was used for categorical variables, and the complex sample generalized linear model was used for continuous variables, to analyze the differences in baseline characteristics based on gender. Complex sample logistic regression was applied to identify the factors associated with adolescents’ mental health. In addition, complex sample multiple logistic regression analyses were used to estimate the adjusted odds ratio (a-OR) and 95% confidence interval (CI) of weight status associated with participants’ mental health after adjusting for possible confounders, such as age, perceived health status, intention to control weight, household income, area of residency, and daily calorie intake. Data were presented as weighted percentages and standard errors (SEs) for categorical variables and as weighted mean ± SE for continuous variables. All analyses were conducted with SPSS software version 24.0 (IBM Co., Armonk, NY, USA). The level of statistical significance was set at a *p*-value of <0.05.

### 2.4. Ethics Statements

All the survey procedures and protocols were approved by the institutional review board of the KCDC, and written informed consent was obtained from all participants and/or their guardians for the survey. In addition, this study was conducted in accordance with the Declaration of Helsinki and approved by the institutional review board of Kangwon National University Hospital, and informed consent was waived, as this study performed a secondary analysis of anonymized open-source data (IRB No. KNUH-2021-06-007).

## 3. Results

### 3.1. General Characteristics of the Study Population

A total of 6264 adolescents, aged between 12 and 18 years, completed the survey. After excluding 452 who did not complete the mental health questions, a total of 5683 adolescents were included in our final analysis. The participants were 53.1% boys and 46.9% girls, and their mean age was 15.1 years old. Among the study population, 20.8% were found to be overweight or obese. The participants’ socio-demographic characteristics are presented in Table 1. 

### 3.2. Body Weight Status of the Study Population According to Sex

Among the study population, the prevalence of obesity was 12.3% of boys and 11.1% of girls during the period from 2010 to 2019. A higher prevalence of being overweight or obese was observed in boys (21.5%) compared to girls (20.2%).

When we further categorized the participants’ self-perceived weight status into three groups (“thin”, “average”, and “fat”) for analysis, approximately one-third of the study population considered themselves to be “fat”. In particular, girls were more likely to perceive themselves as “fat” than boys (35.6% vs. 30.1%), but the difference was not statistically significant. Meanwhile, boys were more likely to consider themselves to be “thin” than girls, with statistical significance [OR, 2.884; 95% CI, 2.435–3.416; *p* < 0.001]. The distribution of self-perceived weight status according to actual weight status (represented by BMI) is shown in Figure 1.

According to the misperception of their weight status, a discrepancy between participants’ actual and self-perceived weight was found in about one-third of the study population. Boys were found to be more likely to underestimate their body weight than girls [OR, 1.640; 95% CI, 1.321–2.035; *p* < 0.001], and girls tended to overestimate their weight status compared to boys [OR, 1.465; 95% CI, 1.240–1.732; *p* < 0.001]. Girls were also found to have attempted to control their weight more often than boys [OR. 2.15; 95% CI, 1.876–2.465; *p* < 0.001].

### 3.3. Health Condition of the Study Population

Among the study population, we found that mental health problems were more prevalent in girls than in boys, with statistical significance. We discovered that 9.1% of the participants had experienced a depressed mood in the preceding two weeks, with a higher prevalence in girls (11.1%) than in boys (7.4%) (*p* < 0.001). The prevalence of reporting feeling stressed in their daily life was 25.7% among all participants and was found to be higher in girls (29.5%) than in boys (22.4%) (*p* < 0.001). We determined that 7.4% of the participants had experienced suicidal ideation in the past year (4.8% of boys and 10.4% of girls, *p* < 0.001). During 2010–2019, no statistically significant trend was observed in the prevalence of depressed mood and perceived stress. However, the prevalence of suicidal ideation statistically significantly decreased over the years (12.5% in 2010 vs. 3.0% in 2019).

Regarding their general health status, we found that boys were more likely than girls to perceive their health as “good or very good”, with statistical significance [OR, 1.201; 95% CI, 1.041–1.386; *p* = 0.012]. According to the unadjusted relationship between perceived general health status and mental health, we determined that, for both sexes, participants who perceived their health status as “bad or very bad” showed a statistically significant prevalence of having experienced a depressed mood, perceived stress, and suicidal ideation. However, socioeconomic status, represented as household income or the area of residency, was not found to be associated with mental health conditions in either sex. The unadjusted associations of the participants’ characteristics with their mental health conditions are presented in Appendix A.

### 3.4. Association between Body Weight Status and Mental Health Conditions

To determine the relationship between actual and perceived weight status and mental health conditions, we analyzed the a-OR (with 95% Cis) of actual and perceived weight statuses for depressed mood, perceived stress, and suicidal ideation, as presented in Table 2.

We found that actual weight status, represented as BMI, was not significantly associated with mental health conditions in either sex. Meanwhile, girls who perceived their bodies to be “fat” were more likely to have experienced depressed mood (a-OR, 1.963; 95% CI, 1.272–3.030; *p* = 0.002) and perceived stress (a-OR, 1.866; 95% CI, 1.389–2.506; *p* < 0.001) than those who perceived their weight to be average. In addition, girls who overestimated their weight, regardless of their actual body weight, were found to be more likely to have experienced depressed mood (a-OR, 1.969; 95% CI, 1.359–2.852; *p* < 0.001) and perceived stress (a-OR, 1.821; 95% CI, 1.376–2.409; *p* < 0.001) than those who accurately matched their perceived weight with their actual body weight. The boys who perceived their weight status as “thin” were much more likely to have experienced suicidal ideation than those who perceived their weight to be average (a-OR, 1.995; 95% CI, 1.143–3.485; *p* = 0.015). 

### 3.5. Subgroup Analysis of the Characteristics Associated with Mental Health Conditions among Obese Adolescents

We performed a subgroup analysis of the data from the overweight/obese participants to determine if there were any specific characteristics associated with their mental health status. A total of 6.8% of the obese group considered their weight status as average or “thin.” In particular, 43.2% of these obese participants misperceived their weight status, with 40.3% underestimating their weight status compared to their actual body weight (Table 1). In addition, when compared with normal/underweight adolescents, overweight/obese participants were found to be more likely to have attempted to control their weight in the past year and were more likely to perceive their general health status as “bad or very bad”, with statistical significance.

In particular, there were no associations between weight perception and mental health conditions among the obese group (Table 3). In other words, the obese participants who perceived their weight as “fat”, in concordance with their actual body weight, had low odds of experiencing mental health problems. The exception was obese girls who perceived themselves as “fat”, who had high odds of experiencing perceived stress and suicidal ideation; however, this difference was not statistically significant.

Meanwhile, there was a statistically significant association between perceived general health status and mental health conditions among obese adolescents. In other words, obese girls who perceived their general health as “bad or very bad” were more likely to have experienced depressed mood, perceived stress, and suicidal ideation, while obese boys with perceived bad health status were more likely to have experienced suicidal ideation. 

## 4. Discussion

Using a large, nationally representative sample, we investigated the relationships between weight status (actual, perceived, or misperceived) and mental health conditions (depressed mood, perceived stress, and suicidal ideation) amongst the general adolescent population. Our study indicates that actual weight status, represented by BMI, was not associated with the existence of mental health conditions in either of the sexes. Meanwhile, girls who perceived themselves to be overweight or who had an overestimated misperception and boys who perceived themselves to be underweight were more likely to experience mental health problems than were participants who perceived themselves to be of average weight or who perceived their weight accurately. However, among obese adolescents, perceptions of their general health status, rather than perceptions or misperceptions about their weight status, were more likely to be associated with mental health conditions.

According to the World Health Organization, one in seven 10 to 19-year-olds experience mental health disorders, accounting for 13% of the global burden of diseases [5,25]. As up to 50% of all mental illnesses present before the age of 14 years and persist throughout adult life, resulting in long-term morbidity, global preventative measures to promote good mental health in adolescents and to reduce the social burden on society are ongoing [25]. In a study of the United States population, the prevalence of major depressive episodes among adolescents aged 12–17 years notably increased from 8.7 to 11.3% between 2005 and 2014 [26]. In our cross-sectional study, the overall prevalence of depressive episodes among Korean adolescents in the past 10 years was 9.1%, with girls being at higher risk than boys, and a quarter of the participants had experienced moderate or severe stress. In a recent study of Korean adolescents, stress caused by conflicts with peers was found to be the number one stressor linked with depressed mood and suicidal ideation, followed by family circumstances [27]. Conflicts with peers are greatly influenced by the appearance of adolescents, who often develop their relationships by communicating about their appearance [28]. Therefore, considering their characteristics, we aimed to report which perceptions about their appearance, such as those related to their actual body weight or perceived body image, could affect mental health in adolescents.

Previous studies have already established the association between actual body weight, represented by BMI, and mental well-being in adults. However, the results from these studies were inconsistent with regard to participants’ sex, age, and ethnicity [29,30]. A meta-analysis of 183 adults, including an Asian population, revealed that both being underweight and being obese were associated with a risk of depression in both sexes, while the association of depression with being overweight differed by sex [15]. In particular, the prevalence of depression among adults has been suggested to be a U-shaped trend, especially when we consider categorized BMI. Depression has been observed to be more prevalent among underweight and obese individuals than among normal-weight or overweight individuals [31]. Several studies focusing on Korean adults and using nationally representative and large, population-based data also support the previous results indicating a U–shaped association between being underweight or severely obese and a high risk of depression or depressive symptoms [32,33,34,35]. However, we did not observe this U-shaped association between actual body weight and depressed mood in Korean adolescents.

Meanwhile, several studies have highlighted the role of self-perceived body weight and how it may influence mental health outcomes, independent of actual weight status. According to several nationwide, representative studies including large sample sizes with diverse ethnic groups, perceived weight—rather than currently measured body weight—was more strongly associated with depression in adults [17,36]. In particular, women who perceived themselves to be overweight/obese and men who perceived themselves as underweight were more likely to experience depression. In studies where the participants were Korean adults, weight perception was also significantly associated with an increased risk of psychological distress, such as depressive symptoms, stress recognition, and suicidal ideation [37,38,39]. Similarly, women who perceived themselves as overweight, including those within the normal BMI group, reported experiencing more depressive symptoms, severe stress, and suicidal thoughts. However, the relationship between perceiving oneself as underweight and depression among men, which has been reported in Western studies, has not been found in studies on Korean men.

The manner in which a child or adolescent perceives their weight status and body image is an important factor that may impact their mental health, with numerous studies revealing the positive association between perceived weight and mental health status, regardless of actual weight. In particular, the perception of being overweight has been shown to be associated with psychological problems, such as major depression, suicidal ideation, and self-reported psychosomatic complaints, when compared with individuals who perceive their body weight to be average [40,41]. In a large, population-based study of Chinese adolescents, perceptions of either being underweight or overweight were found to be related to mental health problems, regardless of BMI [42]. In contrast to previous studies indicating inconsistent results in terms of the gender differences in outcomes, we found clear gender differences in the relationships between perceived weight status and mental health. In particular, girls that perceived themselves to be overweight and boys who perceived themselves to be underweight were at high risk of experiencing mental health problems, which is consistent with several studies of adult samples.

A few studies have proposed that adolescents whose perceived weight does not match their actual body weight experience worse health outcomes than those whose weight status matches their perceptions [43,44]. Their results indicated that normal-weight boys who perceived themselves to be underweight, normal-weight girls who perceived themselves to be overweight, and obese boys and girls who perceived themselves to be overweight were at greater risk of experiencing negative mental health outcomes than individuals who perceived their weight to be normal or average. Our study was designed to explore whether the discrepancy between actual and perceived weight status had an effect on mental health by dividing discordant subjects into two subgroups, namely, an underestimated and an overestimated group. We conclude that overestimated misperceptions were crucial in determining female adolescents’ mental health. This means that special attention should be paid to underweight girls who perceive themselves to be of average weight, as well as to girls who perceive themselves to be obese, as they all fall within the overestimated misperception group. We suggest that even adolescents within a normal weight range should be cautious, as misperceptions of being “fat” can negatively affect adolescents’ mental well-being.

With regard to the findings showing that obese children and adolescents are prone to multiple psychosocial problems, studies on the relationships between childhood obesity and mental health have shown inconsistent results [12,13,14]. In a recent meta-analysis, obese but not overweight children and adolescents were found to be at a significantly higher risk of developing major depressive disorder, not a clinically depressive mood, than their healthy-weight peers [45]. Despite extensive evidence pertaining to the relationship between childhood obesity and mental health, the current research does not provide an understanding of the exact mechanisms of the link between the two diseases. Moreover, it remains uncertain whether mental health problems are a cause or a consequence of childhood obesity or whether common factors promote both obesity and psychiatric disturbances in susceptible children and adolescents [10,46].

In this study, however, we did not observe any relationships between childhood obesity and mental health problems when compared with participants who were of a healthy weight, even in the unadjusted analysis. Moreover, the association between perceived or misperceived weight status and mental health status was not determined among obese adolescents. Instead, the participants’ perceptions of their general health was a risk factor affecting their mental health, suggesting that obese adolescents are more likely to have poorer self-rated health and, as a result, worse mental health outcomes. In a recent Western study, overweight/obese Caucasian adolescents who misperceived their weight to be average were less likely to experience depressive symptoms than those adolescents who accurately perceived their weight as overweight/obese [47]. A recent model suggests that not recognizing that one is overweight may be associated with more favorable physical and mental health outcomes [48]. This model proposes that an individual’s perceptions of being overweight may trigger social rejection concerns and the internalization of weight stigma, which in turn may induce psychological distress and negatively impact health-promoting behaviors. However, other studies have established that obesity is associated with poor self-rated health, life dissatisfaction, mental health problems, and stress, which are risk factors for negative mental health outcomes [27,41]. Therefore, it is difficult to determine whether there is a direct association between mental health conditions and the perception of weight status among obese adolescents through cross-sectional studies, as there are many risk factors that may affect their mental health, including self-perceived health condition and self-rated life satisfaction. Therefore, further research is needed to determine this association after controlling for the possible confounders.

In contrast to previously published studies of Korean adolescents using nationwide surveys, the strength of the present study lies in the fact that we analyzed not only depressive symptoms but also perceived stress and suicidal ideation as representative of the mental health conditions experienced by Korean adolescents over a period of 10 years [49,50]. In addition, the definition of actual body weight status was more accurate in our study, as it was based on the Korean National Growth Chart, in contrast with previous studies that have used the adult-focused BMI classification. Furthermore, the prevalence of mental health disorders was sex-dependent, and we analyzed the study population according to their sex. We also performed a subgroup analysis on obese adolescents. Nevertheless, this study has some limitations. First, the survey used self-reported data on mental health conditions that were assessed symptomatically without using international diagnostic criteria for depression. Second, this study used a cross-sectional design, which restricts our ability to accurately identify the causality between weight status and mental health conditions. Therefore, we recommend that further longitudinal studies be conducted in the future in order to reveal the causality more effectively.

## 5. Conclusions

The findings of this study have practical implications for healthcare providers and parents of adolescents, as perceived body weight plays a more important role than actual body weight in the improving adolescents’ mental health and well-being. We suggest that there is a need to assess adolescents’ perceptions of their body image and weight-related attitudes, particularly with regard to their impact on mental health. Once assessed, we can attempt to modify these perceptions and attitudes, as early intervention is required to promote the mental health of Korean adolescents.

In terms of future research, this study provides a basis for further exploration of the relationship between weight-related attitudes and cultural background to better understand the generalizability of these findings. It also highlights the need for research on interventions that can help adolescents develop more accurate perceptions of their weight status.

## Figures and Tables

**Figure 1 children-10-00620-f001:**
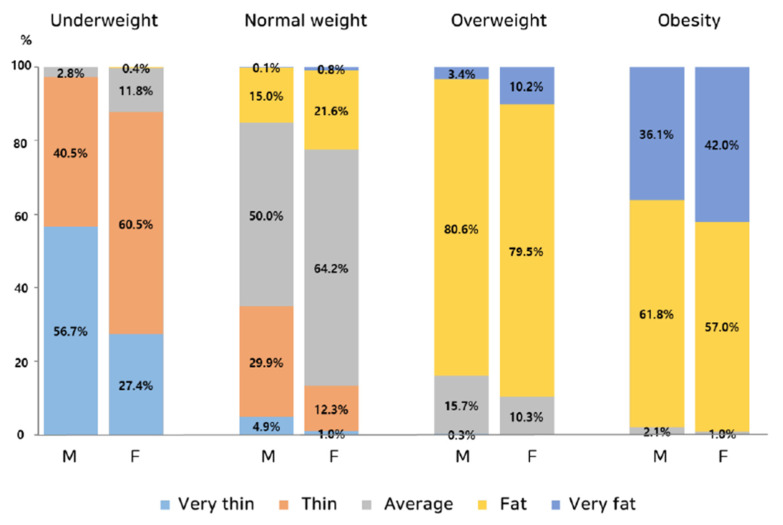
Distribution of self-perceived weight status according to actual weight status (represented by BMI): underweight (below the 5th percentile), normal (from the 5th to the 85th percentile), overweight (from the 85th to the 95th percentile), and obesity (above the 95th percentile); M, male; F, female.

**Table 1 children-10-00620-t001:** Socio-demographic characteristics and health conditions of the study population.

Variables	Total (*n* = 5683)	Male (*n* = 3013)	Female (*n* = 2670)	*p* Value	Overweight & Obese (*n* = 1186)	*p* Value *
Estimated population (%)	4,050,528 (100%)	2,152,279 (53.1%)	1,898,250 (46.9%)		842,414 (20.8%)	
Age (years)	15.1 ± 0.0	15.2 ± 0.0	15.1 ± 0.0	0.450	15.3 ± 0.1	0.007
Total calorie intake (kcal/d)	2204.5 ± 16.5	2486.2 ± 24.6	1887.4 ± 18.1	<0.001	2129.9 ± 32.5	0.006
Body mass index (%tile)				0.001		
Underweight (<5th)	8.3 (0.5)%	9.9 (0.7)%	6.5 (0.5)%			
Normal (5~85th)	70.8 (0.7)%	68.6 (1.0)%	73.3 (1.0)%			
Overweight (85~95th)	9.1 (0.4)%	9.2 (0.6)%	9.1 (0.6)%			
Obesity (>95th)	11.7 (0.5)%	12.3 (0.7)%	11.1 (0.7)%			
Weight perception				<0.001		<0.001
Very thin	6.0 (0.4)%	9.0 (0.7)%	2.5 (0.3)%		0.1 (0.1)%	
Thin	19.1 (0.6)%	24.6 (0.9)%	13.0 (0.8)%		0.0%	
Average	42.1 (0.8)%	36.2 (1.0)%	48.8 (1.2)%		6.7 (0.8)%	
Fat	27.2 (0.7)%	25.3 (0.9)%	29.3 (1.1)%		68.6 (1.5)%	
Very fat	5.5 (0.4)%	4.8 (0.5)%	6.3 (0.6)%		24.6 (1.5)%	
Weight misperception				<0.001		<0.001
Concordance	71.1 (0.7)%	72.3 (1.0)%	69.8 (1.0)%		56.8 (1.6)%	
Discordance, underestimated	10.5 (0.5)%	12.7 (0.7)%	8.1 (0.6)%		40.3 (1.6)%	
Discordance, overestimated	18.4 (0.6)%	15.0 (0.8)%	22.1 (0.9)%		2.9 (0.5)%	
Intention to control weight				<0.001		<0.001
No	33.0 (0.7)%	40.2 (1.0)%	24.8 (1.0)%		13.8 (1.1)%	
Yes	67.0 (0.7)%	59.8 (1.0)%	75.2 (1.0)%		86.2 (1.1)%	
Perceived health status				0.009		<0.001
Good/Very good	59.1 (0.8)%	61.4 (1.1)%	56.6 (1.1)%		48.2 (1.7)%	
Fair	35.2 (0.8)%	33.2 (1.1)%	37.4 (1.1)%		43.1 (1.7)%	
Bad/Very bad	5.7 (0.4)%	5.4 (0.5)%	6.1 (0.5)%		8.6 (1.0)%	
Depressed mood				<0.001		0.409
No	90.9 (0.5)%	92.6 (0.6)%	88.9 (0.7)%		90.2 (1.0)%	
Yes	9.1 (0.5)%	7.4 (0.6)%	11.1 (0.7)%		9.8 (1.0)%	
Perceived stress				<0.001		0.060
None/Mild	74.3 (0.7)%	77.6 (0.9)%	70.5 (1.0)%		71.9 (1.5)%	
Much/Very much	25.7 (0.7)%	22.4 (0.9)%	29.5 (1.0)%		28.1 (1.5)%	
Suicidal ideation				<0.001		0.360
No	92.6 (0.4)%	95.2 (0.5)%	89.6 (0.7)%		91.9 (0.9)%	
Yes	7.4 (0.4)%	4.8 (0.5)%	10.4 (0.7)%		8.1 (0.9)%	
Household income				0.188		0.846
High	29.4 (1.0)%	29.7 (1.1)%	29.1 (1.3)%		29.7 (1.6)%	
Middle	57.5 (1.0)%	58.1 (1.2)%	56.8 (1.3)%		56.8 (1.7)%	
Low	13.1 (0.7)%	12.2 (0.8)%	14.1 (0.9)%		13.5 (1.3)%	
Area of residency				0.058		0.564
Urban area	84.5 (1.2)%	85.4 (1.2)%	83.4 (1.4)%		85.1 (1.6)%	
Rural area	15.5 (1.2)%	14.6 (1.2)%	16.6 (1.4)%		14.9 (1.6)%	

Values are expressed as weighted percentages (standard error) or weighted mean ± standard error; * comparison between overweight/obese and underweight/normal weight patients.

**Table 2 children-10-00620-t002:** Associations between body weight status and mental health conditions.

	Depressed Mood		Perceived Stress		Suicidal Ideation	
	Male	Female	Male	Female	Male	Female
Body mass index ^1^						
Underweight	1.119 (0.651–1.923)	1.103 (0.612–1.987)	0.751 (0.530–1.065)	1.362 (0.898–2.067)	1.137 (0.575–2.249)	0.701 (0.342–1.439)
Overweight	1.197 (0.674–2.126)	1.166 (0.717–1.896)	1.052 (0.709–1.561)	1.002 (0.691–1.453)	1.195 (0.631–2.263)	1.164 (0.662–2.046)
Obesity	0.964 (0.513–1.814)	0.799 (0.473–1.349)	0.997 (0.706–1.407)	1.070 (0.765–1.498)	1.032 (0.539–1.974)	0.887 (0.545–1.444)
Weight perception ^2^						
Thin/Very thin	0.863 (0.528–1.409)	1.159 (0.671–2.001)	1.138 (0.854–1.514)	0.980 (0.676–1.421)	**1.995** * (**1.143–3.485**)	1.532 (0.868–2.702)
Fat/Very fat	1.207 (0.660–2.207)	**1.963** ^†^ (**1.272–3.030**)	1.448 (0.992–2.114)	**1.866** ^‡^ (**1.389–2.506**)	1.193 (0.532–2.676)	1.505 (0.978–2.315)
Weight misperception ^2^						
Discordance, underestimated	1.543 (0.860–2.767)	1.545 (0.782–3.052)	1.391 (0.931–2.079)	0.852 (0.512–1.418)	1.329 (0.611–2.891)	0.616 (0.301–1.261)
Discordance, overestimated	1.234 (0.755–2.016)	**1.969** ^‡^ (**1.359–2.852**)	1.232 (0.879–1.725)	**1.821** ^‡^ (**1.376–2.409**)	1.074 (0.599–1.926)	1.345 (0.896–2.018)

Values are presented as adjusted odds ratios (95% confidence interval); ^1^ adjusted for age, intention to control weight, perceived health status, household income, area of residency, total calorie intake; ^2^ adjusted for age, categorized body mass index, intention to control weight, perceived health status, household income, area of residency, total calorie intake; the reference was normal, average, and concordance, respectively. The bold typeface indicates statistical significance; * *p*-value <0.02, ^†^
*p*-value < 0.005, ^‡^
*p*-value < 0.001.

**Table 3 children-10-00620-t003:** Associations between weight status and mental health of overweight and obese adolescents.

	Depressed Mood		Perceived Stress		Suicidal Ideation	
	Male	Female	Male	Female	Male	Female
Weight perception ^1^						
Average	1.000	1.000	1.000	1.000	1.000	1.000
Fat/Very fat	0.744 (0.205–2.702)	0.876 (0.229–3.345)	0.988 (0.452–2.160)	2.109 (0.807–5.510)	0.604 (0.157–2.323)	6.179 (0.804–47.509)
Weight misperception ^1^						
Discordance, underestimated	1.357 (0.644–2.859)	1.157 (0.569–2.354)	1.143 (0.727–1.798)	0.745 (0.466–1.191)	1.242 (0.531–2.908)	0.561 (0.263–1.194)
Concordance	1.000	1.000	1.000	1.000	1.000	1.000
Perceived health status ^2^						
Good/Very good	0.901 (0.403–2.015)	0.675 (0.332–1.372)	0.818 (0.511–1.310)	**0.479** ^†^ (**0.298–0.769**)	1.188 (0.433–3.259)	0.653 (0.299–1.424)
Bad/Very bad	0.690 (0.177–2.684)	**3.432** * (**1.190–9.903**)	1.746 (0.819–3.722)	**2974** * (**1.187–7.450**)	**6.977** ^‡^ (**2.402–20.265**)	**5.534** ^†^ (**2.038–15.028**)

Values are presented as adjusted odds ratios (95% confidence interval); ^1^ adjusted for age, intention to control weight, perceived health status, household income, and area of residency; ^2^ adjusted for age, intention to control weight, weight perception, household income, and area of residency, with fair as a reference. The bold typeface indicates statistical significance; * *p*-value < 0.05, ^†^
*p*-value < 0.005, ^‡^
*p*-value < 0.001.

## Data Availability

KNHANES data are publicly available and can be accessed online (http://knhanes.kdca.go.kr/knhanes/eng/index.do), (accessed on 12 August 2022).

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
