# Peer review of "Association between Weight Status and Mental Health among Korean Adolescents: A Nationwide Cross-Sectional Study"

_children, 2023, doi:10.3390/children10040620_

Round 1

Reviewer 1 Report

Title: Association between weight status and mental health among Korean adolescents: A nationwide cross-sectional study

To Authors:

I appreciate this manuscript's topic and provide valuable information about the relationship between adolescents’ weight status and their mental health among Korean adolescents using the public data. The premise behind this manuscript is robust and would be of interest to the readers of the journal; thus, this manuscript can be reconsidered for publication pending major revisions. There are a few points that could be further developed or improved upon listed below.

Abstract

1.      Please add the year of the public data.

2.      Since this study applied group differences, please add the detailed information about the subgroups, such as mean of age, % of boys vs. girls, overweight/obese adolescents vs. normal vs. thin in actual/perceived. In addition, it would be helpful to know the exact percentages of adolescents who experienced depressed mood, stress, or suicidal ideation based on their weight status.

3.      Please add the one sentence of suggestions/implications based on the findings.

Introduction

1.      The mental health issues mentioned in the first paragraph, such as behavioral disorders, eating disorders, and substance use disorders, are not typically considered mental health issues but rather health risk behaviors.

2.      The transition between the first and second paragraphs could be smoother to better connect the topics of mental health and childhood obesity.

3.      The introduction could benefit from a more thorough literature review that highlights the research gaps in previous studies and explains the novelty of the current study. Specifically, it would be helpful to provide more specific information on what this study adds to the existing literature on the relationship between overweight/obesity and mental health in adolescents.

4.      As the study is based primarily on South Korean adolescents, it would be valuable to include information on the cultural backgrounds that may influence the prevalence of mental health issues and overweight/obesity in this population.

5.      Consideration could be given to the influence of cultural norms on body image distortion (related to misperceptions of their weight), particularly in Eastern Asian countries compared to Western countries.

6.      It would also be helpful to clearly state the research questions in the study, as this will help readers understand the scope and direction of the research project.

7.      Given that the findings primarily highlight sex differences between boys and girls, it would be helpful to provide a literature review in the introduction to support these findings.

Results

1.      It would be helpful to include the standard deviation (SD) of the sample's mean age.

2.      As this study analyzed data spanning a 10-year period, it would be valuable to provide information on the trends in mental health prevalence among Korean adolescents from 2010 to 2019 as a point of comparison.

Discussion

1.      Line 286: Did you mean to write "BMI" instead of on “BM”?

2.      While there is evidence to suggest a relationship between actual and perceived weight status, it may be useful to temper the study's conclusion that perceived body weight plays a more important role than actual body weight in mental health outcomes.

3.      Including more practical implications in the conclusion section of the paper would be valuable for readers seeking to apply the study's findings in real-world settings.

I hope this feedback is helpful in improving the overall quality of the manuscript.

Author Response

We appreciate the time and effort that you and the reviews have dedicated to providing your valuable feedback on our manuscript. We are grateful to the reviewers for their insightful comments on our paper. We have been able to incorporate changes to reflect most of the suggestions provided by the reviewers. We have highlighted the changes within the manuscript.

Here is a point-by point response to the reviewers’ comments and concerns.

Reviewer 2 Report

First of all, I have to say that I found this manuscript interesting enough to be considered for publication in this journal. However, before doing so, the authors should take into account a series of considerations:

Abstract

-       It would be advisable to eliminate all those expressions mentioned in personal language such as "We analyzed", "We found" and to be modified by impersonal language.

-       There is no mention of any statistical methodology used in this article, it is necessary to include it.

Introduction

-       In line 28, in the sentence "In addition to the process...", it would be necessary to include a bibliographic reference to verify this information.

-       Perhaps the introduction is too sparse, more information should be included as well as relevant studies on the variables studied in this article.

-       In the objective of the study it is mentioned the following: "...this study aimed to investigate the relationship between actual body weight status or weight perception". In this sense, when is the body weight status variable taken into account and when is the perception of body weight? The connector "or" instead of "and" may lead to doubts for the reader.

Material and methods

-       Inclusion and exclusion criteria for study participants are not included.

Discussion

-       It is essential to include the practical applications of this study as well as its contribution to future research.

Conclusions

-       The conclusions should be in a different section from the discussion, taking into account the requirements and formal peculiarities of the journal.

Finally, the authors' contributions should be reviewed as they do not follow the format of this journal.

Author Response

(The authors gave the same response as above.)

Round 2

Reviewer 1 Report

Thank you for your effort in revising this study based on my previous feedback. The quality of the manuscript has been improved and is ready to be published in this journal.

Lastly, I recommend that you can add the research reference on line 46.

Shen, L., Gu, X., Zhang, T., & Lee, J. (2022). Adolescents’ physical activity and depressive symptoms: A psychosocial mechanism. International Journal of Environmental Research and Public Health19(3), 1276.

Author Response

We appreciate the time and effort you put into providing valuable feedback on our review responses. We are grateful to you confirming the response and thorough review. We have attempted to amend the change to reflect the suggestion. We have highlighted the changes within the manuscript.

Response to the comments by Reviewer 1:

Comment

To Authors:

Thank you for your effort in revising this study based on my previous feedback. The quality of the manuscript has been improved and is ready to be published in this journal.

Response

We appreciate you for the feedback on our review response. We revised the manuscript as you pointed.

Comment #1

I recommend that you can add the research reference on line 46.

Shen, L., Gu, X., Zhang, T., & Lee, J. (2022). Adolescents’ physical activity and depressive symptoms: A psychosocial mechanism. International Journal of Environmental Research and Public Health19(3), 1276.

Response

We added the reference as follows:

  • Page 2, line 45: These factors can make them vulnerable to various mental health issues including risk-taking behaviors, such as depression, anxiety, behavioral disorders, eating disorders, suicide, or substance use disorders [5,6,7].

Sincerely,

Youngha Choi and Jeana Hong

Reviewer 2 Report

The manuscript has been considerably improved and I believe it should be published in this journal

Author Response

We appreciate the time and effort you put into providing valuable feedback on our review responses. We are grateful to you confirming the response and thorough review. Thank you for your consideration. 

Sincerely,

Youngha Choi and Jeana Hong